# Nanocellulose from Spanish Harvesting Residues to Improve the Sustainability and Functionality of Linerboard Recycling Processes

**DOI:** 10.3390/nano12244447

**Published:** 2022-12-14

**Authors:** Jorge De Haro-Niza, Esther Rincón, Zoilo Gonzalez, Eduardo Espinosa, Alejandro Rodríguez

**Affiliations:** 1BioPrEn Group (RNM940), Chemical Engineering Department, Faculty of Science, Instituto Químico para la Energía y el Medioambiente (IQUEMA), Universidad de Córdoba, 14014 Córdoba, Spain; 2Department of Food Science and Technology, Faculty of Veterinary, Universidad de Córdoba, 14014 Córdoba, Spain

**Keywords:** horticultural residues, vine shoots, cellulose nanofibers, linerboard recycling

## Abstract

The hornification processes undergone by the fibers in the paper industry recycling processes lead to the loss of properties of the final products, which exhibit poor mechanical properties. Among the most promising solutions is the reinforcement of secondary fibers with cellulose nanofibers. The present work addresses two important issues: the efficient production of cellulose nanofibers from scarcely exploited agricultural wastes such as horticultural residues and vine shoots, and their application as a reinforcement agent in recycled linerboard recycling processes. The effect of the chemical composition and the pretreatment used on the nanofibrillation efficiency of the fibers was analyzed. Chemical pretreatment allowed a significantly higher nanofibrillated fraction (45–63%) than that produced by mechanical (18–38%), as well as higher specific surface areas (>430 m^2^/g). The application of the nanofibers as a reinforcing agent in the recycled linerboard considerably improved the mechanical properties (improvements of 15% for breaking length, 220–240% for Young’s modulus and 27% for tear index), counteracting the loss of mechanical properties suffered during recycling when using chemically pretreated cellulose nanofibers from horticultural residues and vine shoots. It was concluded that this technology surpasses the mechanical reinforcement produced by conventional mechanical refining used in the industry and extends the number of recycling cycles of the products due to the non-physical modification of the fibers.

## 1. Introduction

The environmental problems faced by society, together with the existing economic challenges, have led industries to adopt more and more measures for the application of the circular economy in their processes as the main axis of sustainable development. The paper and linerboard industry in among these sectors that have adopted policies for the recycling of its products. The main strategy for this purpose is the use of recycled secondary fibers in the production of linerboard. Unfortunately, the products obtained have worse mechanical performance than with the original fibers due to the hornification process [1].

Hornification refers to dry fibers that do not fully reswell in water, forming numerous hydrogen bonds between the microfibrils during drying [2]. This process causes the fiber volume to shrink, causing the hornified recycled paper to have, in addition to low mechanical properties, poor hydration and swelling properties. One way to alleviate this is mechanical beating of the recycled paper. This process restores the swelling capacity of the hornified fibers by increasing their fines content and surface area, which facilitates fiber–fiber bonding. However, after several cycles of mechanical refining, the problems produced in the fibers by the effects of hornification are irreversible [3]. Nevertheless, the use of recycled paper and linerboard produced from secondary fibers saves water and chemicals, as well as reducing air and water pollution compared to the production of new paper or linerboard [2]. It is therefore necessary to find solutions to this deterioration of the fibers that will allow the production of recycled paper with mechanical properties comparable to virgin products. For this purpose, there are different approaches such as the classic treatments applied in the industry as the addition of virgin fiber or chemicals, or more innovative treatments such as enzymatic beating [4], the addition of binders such as cationic starch or nanofillers such as cellulose nanofibers [3,5].

Cellulose nanofibers (CNF) have been used at various stages of paper and linerboard production (refining, forming, pressing, and drying) as a reinforcing agent, retention system component, printing quality aid, coating binder and barrier agent that controls water vapor and oxygen permeability. It is also used as a substitute for inorganic fillers in paper sheets to maintain the mechanical resistance [6]. There are two main hypotheses regarding the mechanism of action underlying the improvement by adding CNF to the fibrous matrix. The first one is that the addition of CNF generates an embedded network between the macroscopic fibers, which increases the load-bearing capacity of the paper or linerboard. The second proposed mechanism is that CNF promotes adhesion by bridging adjacent fibers favoring fiber–fiber bonding and thus increasing the bonding area. The addition of CNF not only improves the mechanical properties of the final products but also decreases their thickness, reducing their density and porosity [1]. The use of CNF as a reinforcing agent in paper and linerboard has already been studied by numerous authors. The addition of CNF improves the mechanical strength and durability of papers [7], stabilizes flocs and minimizes overdosing effects [8,9]. In the case of linerboard, mechanical properties are also considerably improved [10,11]. In this sense, it is more than demonstrated that CNF can be used as bio-based substitutes for the synthetic reinforcing materials currently used in the paper and linerboard industry.

Despite the advantages reported by the addition of CNF in paper slurries, the costs of manufacturing and applying CNF in this type of industry are not completely clear. On the one hand, chemical pretreatment (TEMPO-mediated oxidation) applied to cellulose pulps prior to their passage through the high-pressure homogenizer has been recognized as the most effective pretreatment. However, the cost of the chemical reagents involved is high, making this process difficult to scale up to industry. On the other hand, it has been shown that the presence of residual lignin and hemicelluloses in the fibers facilitates the mechanical dismantling of the cellulose fiber structure, improves the dehydration of the suspensions, and reduces their hydrophilicity, so it might not be necessary to use CNF, prior bleaching treatment of the cellulose pulp, but rather lignocellulose nanofibers (LCNF) could be used [12]. In any case, the improvement of mechanical properties in the final paper and linerboard will be obtained by the addition of (L)CNF with specific characteristics, such as high specific surface area.

To all this must be added the raw material from which the (L)CNF is obtained. Normally, wood raw materials are used but, in recent years, the use of agricultural residues for this purpose has aroused great interest. This is due to the fact that agricultural and agri-food residues are a highly available and low-cost resource [13]. In 2021, fruit and vegetable production in Europe was 105 million tons, with Spain being the largest producer (26%). Fruit and vegetable production in Spain is led by Andalusia whose production represents 34% of the total [14]. If it is estimated that 15–40% of a crop ends up as agricultural residues [15], fruit and vegetable production leave a large amount of waste that must be managed. In Spain, the law 7/2022 on waste and contaminated soils for circular economy prohibits the burning of this residues and requires their reuse, recycling, and management [16]. Therefore, these agricultural residues, as lignocellulosic materials, could be used to produce (L)CNF.

In this context, the aim of the present study was to study the viability of two different Spanish agricultural residues to produce LCNF and CNF, specifically a mixture of tomato, eggplant and bell pepper residues, and vine shoot residues, and subsequent application as reinforcement in recycled linerboard, as a solution for the paper and linerboard industry. This study evaluated two different pretreatments to produce the nanofibers as well as the effect of residual lignin for this purpose.

## 2. Materials and Methods

### 2.1. Materials

Bell pepper (*Capsicum annuum*), tomato (*Solanum lycopersicum*) and eggplant (*Solanum melongena*) horticultural residues were supplied by Cooperativas Agrícolas de Almería (Andalucía, Spain) as a vegetable mixture of identical proportions. The vine shoots (*Vitis vinifera*) were supplied by Instituto de Investigación y Formación Agraria y Pesquera (IFAPA) from an ecological crop in Cabra (Andalucía, Spain). Prior to all the experimental work, the biomass was stored, dried and ground to reduce the particle size to 0.25–0.40 mm according to the T264-cm-88 standard [17]. The recycled linerboard fibers were supplied from Smurfit Kappa container S.L. (Mengibar, Jaén, Spain).

The content of α-cellulose, holocellulose, lignin, ashes, ethanol extractable and hot water solubility was determined according to TAPPI standards T-9m54, T-222, T-203os-61, T-211, T-204, T-257 and T-212, respectively [17].

Chemical reagents used in this work consisted of sodium hydroxide (NaOH, ≥97% purity) from PanReac AppliChem ITW Reagents, sodium chlorite (NaClO_2_, >80% purity), sodium hypochlorite (NaClO, 80% purity), and sodium bromide (NaBr, 99% purity) supplied by Honeywell Fluka^TM^, acetic acid (≥99.7% purity) supplied by ACS reagent, acetone and ethanol from Alcoholes del Sur S.A., 2,2,6,6-tetramethy-1-piperidinyloxy (TEMPO, free radical, 98% purity) and colloidal silica (LUDOX^®^ HS-40) supplied by Sigma Aldrich (St. Louis, MO, USA), and cationic starch (Vector SC 20157) supplied from Roquette (Lestrem, France).

### 2.2. Production and Characterization of Cellulose Pulps

Cellulose pulps, both from horticultural residues (HR) and vine shoots (VS), were obtained following the same semi-chemical process. For this purpose, the raw material was pulped in a 15 L batch reactor under an environmentally-friendly soda process conditions (100 °C, 150 min, 7% o.d.m. NaOH, liquid/solid ratio 10:1), following the protocol reported in previous investigations [3]. The pulping yield was calculated following the equation proposed in previous investigations [18]. These unbleached cellulose pulps (UBP) were submitted to a bleaching treatment using 0.3 g NaClO_2_/g pulp in a 0.3% pulp suspension in water at 80 °C for 3 h. Finally, the pulp suspension was filtered and washed with acetone and water to obtain bleached cellulose pulp (BP). Cellulose pulps were characterized following TAPPI standards as described for the raw materials.

### 2.3. Preparation and Characterization of (L)CNF

Both unbleached (UBP) and bleached (BP) pulps were used to produce LCNF and CNF, respectively. Before (L)CNF production, two different pretreatments were assessed: mechanical and chemical pretreatment. In the case of mechanical pretreatment, the pulp was refined in a PFI mill until a Schopper–Riegler degree (°SR) of 90 (ISO 5264-2:2002) was achieved. Then, a suspension was prepared with the pretreated fibers at 1.5% *wt*. and passed through a GEA Niro Panda Plus 2000 high-pressure homogenizer (HPH), following optimized sequence [3]. The chemical pretreatment consisted of a TEMPO-mediated oxidation following the procedures and proportions described by Besbes et al. [19]. The reaction was conducted at pH 10.2 and started with the addition of NaClO solution to achieve an oxidative power of 5 mmols NaClO/g fiber. Once the addition of NaClO was finished, a 0.5 M NaOH solution was used to maintain the pH. Finally, the pretreatment was finished when the pH remained stable without NaOH addition, and a 1.5% TEMPO-oxidized fiber suspension was subjected to the same HPH process as previously mentioned. Table 1 summarizes the nomenclature of the nanofibers obtained in this work.

The nanofibrillated fraction for each nanofiber suspension was determined by centrifuging a 0.1% nanofiber suspension for 20 min at 10,000 rpm [19]. Likewise, the optical transmittance was also determined in a 0.1% nanofiber suspension by measuring the absorbance at 800 nm in a Perkin–Elmer spectrometer UV/Vis Lambda-25 (Whaltham, MA, USA). As an indirect measurement of the specific surface area of the nanofibers, cationic demand (CD) and carboxyl content (CC) were determined. Considering the stoichiometric relationship between hydroxyl and carboxyl groups with poly-DACMAC, the specific surface area (*σ*) and diameter (*d*) of the nanofibers were estimated, following previously published methodologies [3], according to Equations (1) and (2), respectively.
(1)$$\sigma_{nanofibers}\left(m^{2}/g\right)=\left(CD-CC\right)\cdot \sigma_{DADMAC}$$
(2)$$d\;\left(\mathrm{nm}\right)=\frac{4}{\sigma_{nanofibers}\cdot 1600\cdot 10^{3}}$$

The degree of polymerization (DP) was determined from the intrinsic viscosity measurements, according to UNE 57-039-92 [20]. The DP data were used to calculate the length of the fibers according to the Equation (3), proposed by Shinoda et al. [21].
(3)Length (nm)=4.286·DP−757

The chemical structure of the nanofibers was analyzed by Fourier Transform Infrared (FTIR) spectroscopy using a Spectrum Two^TM^ instrument (Perkin Elmer, Waltham, MA, USA) by the attenuated total reflectance (ATR) technique, recording the spectra from 4000 to 400 cm^−1^ with a resolution of 4 cm^−1^ collecting a total of 20 scans. The crystallinity of the samples was determined from the X-Ray diffraction (XRD) patterns which were recorded using a Bruker D8 Discover with a monochromatic source CuKα1 over an angular range of 10–80° at a scan speed of 0.025°/s in reflection mode. The crystallinity index (CI) was calculated from the intensity of the peak corresponding to (200) plane of cellulose crystalline and the minimum intensity between the peaks associated to (200) and (110) planes following the equations described by Segal et al. [22].

### 2.4. Production and Evaluation of Reinforced-Recycled Linerboard Sheets

The reinforcing effect of nanofibers addition on recycled linerboard was studied by adding them in the formulation of linerboard sheets at 3% on fiber dry weight. For this purpose, 30 g of recycled linerboard fibers were disintegrated in a pulp disintegrator for 30 min at 3000 rpm. The nanofibers were then added to the linerboard suspension and disintegrated for 1 h at 3000 rpm. Subsequently, 0.5% cationic starch and 0.8% colloidal silica were added (on fiber dry basis). All this mixture was kept under constant agitation for 30 min before linerboard sheet formation in an ENJO-F-39-71 sheet former, according to TAPPI T 205 sp-02 standard [17], with a basis weight of 60 g/m^2^. A batch of 10 linerboard sheets was produced for each type of nanofibers. Finally, the linerboard sheets were conditioned at 25 °C and 50% RH prior to the mechanical characterization. In order to compare the reinforcement of the nanofibers of this work with the methods applied in the industry such as the mechanical beating, a batch of linerboard sheets formed by linerboard refined fibers in a PFI beater at 2000 and 3000 rev was formed.

The mechanical characterization of the obtained linerboard sheets was carried out by determining the breaking length and Young’s modulus, burst index, and tear index according to TAPPI standards T-494-om 01, T-403-om 97 and T-414-om98, respectively [17].

## 3. Results and Discussion

### 3.1. Nanofibrilation Suitability Study of Agricultural Residues Fibers

Among the most important factors underlying the valorization of agricultural residues is their chemical composition, which will determine their suitability for the production of cellulose pulps. The chemical composition of horticultural residues and vine shoots as well unbleached and bleached cellulose pulps obtained is displayed in Table 2.

For both residues, the α-cellulose content was increased after the pulping process (50.45 and 51.89% for UBPs) indicating the suitability of the chosen treatment conditions. Additionally, the bleaching process employed in both UBPs was effective in removing lignin, which decreased to less than 8%, further enriching the pulps in α-cellulose content (more than 60%). The α-cellulose content determined in HR and VS was similar to other agricultural residues used successfully in the production of cellulose nanofibers such as olive prunings (41%) [23], and cereal straws (34–40%) [24]. Besides cellulose, the presence of lignin and hemicelluloses in the fibers gives them certain properties that faclitate mechanical delamination [12]. The presence of these components has a variable effect depending on the type of pretreatment used. For TEMPO-mediated oxidation pretreatment, the presence of lignin leads to excessive consumption of chemical reagents and competes with the fibers themselves [25], and the presence of hemicelluloses decreases the accessibility to the inner surface of the fibers by decreasing the carboxyl content of the pretreated fibers. However, when a mechanical pretreatment is applied, hemicelluloses facilitate nanofibrillation by inhibiting the coalescence of the microfibrils through hydrogen bonds, increasing the stability of the fibrils in the aqueous suspension [26]. In addition, for mechanical pretreatment, a high lignin content (more than 20%) reduces the efficiency of nanofibrillation but a residual content can prevent previously broken covalent bonds from re-forming [1]. After the bleaching process, the lignin content became residual, maintaining the hemicellulose content practically intact, making it possible to study the effect of lignin on the efficiency of nanofibrillation on the different pretreatments.

For the different cellulose pulps from both agricultural residues, two different pretreatments were evaluated: mechanical pretreatment and TEMPO-mediated oxidation (Table 1). The obtained nanofibers were characterized in terms of nanofibrillated fraction (η), optical transmittance (T_800_), cationic demand (CD), carboxyl content (CC) and morphology (Table 3).

Generally, it was observed that the nanofibrillated fraction was much higher when chemical pretreatment was performed (45–63%) than mechanical pretreatment (18–38%). This is because the high surface charge conferred to the fiber during pretreatment due to the oxidation of the primary C6 alcohol of the cellulose into carboxyl groups facilitates its subsequent delamination during the nanofibrillation process [1]. Optical transmittance (T_800_) is an indirect indicator of nanofibrillated fraction resulting in higher light scattering (lower transmittance) as the fraction of nanofibers decreases. This parameter highlights once again the higher nanofibrillation produced in chemically pretreated fibers. Regarding the effect of lignin on the nanofibrillation efficiency, it was evidenced that for both pretreatments, a high lignin content results in a lower nanofibrillation of the fibers. Indeed, this is due to the secondary lignin oxidation reactions produced by the reagents during TEMPO-mediated oxidation that decreases the reagent available for cellulose oxidation, and by the higher bonding of the fiber-forming components due to the presence of lignin that hinders fibrillation during mechanical pretreatment [25]. Slight differences were observed between the raw materials studied, and higher yields were obtained in HR, possibly due to the higher hemicellulose content present. The nanofibrillated fractions obtained were similar to those reported in the literature for other agricultural residues [24].

A parameter directly related to the specific surface area is the CD, since it determines the interaction capacity of the anionic surface of the nanofibers with the surrounding area. The CD of the nanofibers is influenced by the CC and the specific surface area. As expected, it was observed that the nanofibers obtained by TEMPO-mediated oxidation showed a much higher CD (620–1230 µeq·g/g) compared to the mechanical ones (301–689 µeq·g/g) due to the conversion of alcohols to carboxyls. The effect of lignin on TEMPO oxidization efficiency is clearly seen in the lower carboxyl content and lower CD of LCNF compared to CNF. As mentioned above, this fact can be explained by the lower specificity of the cellulose oxidation reaction for secondary lignin oxidation reactions. As for those subjected to mechanical pretreatment, these differences are driven by the lower fibrillation performance of LCNF. The CC followed the same trend of results. The bleaching process in the pulp removes the lignin, leaving the fibers more accessible for TEMPO-mediated oxidation, hence the highest CC values occur in TCNF [7], thus obtaining the lowest nanofiber diameters with the highest specific surface areas. For other agricultural residues such as cereal straws (wheat, rice, barley, and oats), similar values have been reported for the properties of nanofibers in which their characteristics have been compared as a function of the pretreatment applied and the presence or absence of lignin in them [27]. The strongest depolymerization occurs in TCNF as a result of excessive oxidation of the amorphous regions of the cellulose resulting in the shortest fiber lengths, as evidenced in this work. A large specific surface area and a nanosized diameter value in the nanofiber allows a greater capacity of interaction with the surrounding fibers, thus increasing the fiber–fiber adhesion and favoring the reinforcing effect of the final product.

The chemical structure of the different nanofibers was also analyzed by FTIR-ATR spectroscopy (Figure 1). The spectra of both raw materials for all types of nanofibers showed the typical band at 3330 cm^−1^ due to the aromatic and aliphatic O-H stretching vibrations [28]. The peak at 2895 cm^−1^ is related to the stretching of the C-H and CH_2_ bonds of the aromatic structures of cellulose and hemicelluloses [29]. The band at 1603 cm^−1^, is assigned to carboxyl groups stretching. This band sharpened to a more intense peak in the case of chemically pretreated fibers, resulting from asymmetric stretching of the carboxyl groups present due to TEMPO-mediated oxidation [30]. In addition, higher intensity is shown in the chemically pretreated lignin-free samples due to the higher pretreatment efficiency. The peak at 1504 cm^−1^ corresponds to the stretching of the aromatic ring of the lignin, hence it is only observed in LCNF and TLCNF spectra, indicating the success in the bleaching process for lignin removal [31]. The peak observed at 1428 cm^−1^ is attributed to the deformation of the symmetric C-H and CH_2_ bond of cellulose and hemicelluloses. The peaks at 1105 and 1030 cm^−1^ are associated with the stretching of the CH-OH and CH_2_-OH bonds of the secondary and primary alcohols, respectively, in the carbohydrates present. Finally, the peak at 896 cm^−1^ corresponds to the β-glycosidic bonds of cellulose and hemicelluloses [29].

The X-ray diffraction patterns as well as the crystallinity index (CI) data of all samples are also shown in Figure 1. In all cases, the diffractograms showed the expected peaks at 2θ = 15.7° and 2θ = 22.3°, corresponding to the (110) and (200) crystalline planes typical of the crystal structure of cellulose I [24]. In the case of VS nanofibers, a small crystalline peak at 2θ = 14.9° was also observed, as a result of the superimposition of the crystalline planes (1–10) and (110) [1]. The diffractograms show how the process used to obtain the nanofibers had a significant effect on the CI, as well as the raw material from which they were obtained. In the case of HR nanofibers, the CI was affected by the pretreatment used to obtain the nanofibers. Although TEMPO-mediated oxidation degrades the amorphous regions of the cellulose while maintaining the crystalline ones, a decrease in the CI of the chemically pretreated HR nanofibers was observed. As an example, HR-CNF showed a CI of 66.49%, while that of HR-TCNF was 50.87%. The explanation behind this phenomenon is the high pressure reached during the production of the nanofibers by HPH which produces a cleavage of the oxidized crystalline regions with TEMPO [32]. In the case of mechanical pretreatment, the crystalline regions are not as weakened as in TEMPO-mediated oxidation, hence the slightly higher CI. The chemical composition of the starting cellulose pulp also had some effect on CI. Thus, HR-LCNF had a slightly lower CI than HR-CNF due to the higher hemicelluloses and lignin content (amorphous component) of the former (Table 2). In the case of VS nanofibers, this same increase in CI was observed for nanofibers with lower lignin and hemicellulose content (VS-CNF and VS-TCNF). The CI values obtained in this study for HR and VS nanofibers were similar to those obtained for other agricultural residues such as cereal straws (wheat, barley corn and oats) with CI in the range of 44–59% [27] or orange tree prunings which exhibited a CI of 58% for TLCNFs [1].

### 3.2. Production and Characterization of the Reinforced Recycled Linerboard

In general, good mechanical performance of linerboard sheets is affected by fiber strength, specific bond area, forming process and residual stress distribution. Any treatment that improves one of these characteristics will favorably affect the mechanical properties of the final products [33]. Therefore, reinforced linerboard sheets with each of these nanofibers were produced. The results were compared with recycled linerboard (without the addition of nanofibers, named as Ref.) and with linerboard sheets containing fibers subjected to mechanical beating (2000 and 3000 rev), among the classic treatments applied in the industry to alleviate the aforementioned problems. These comparisons between the materials were made in terms of mechanical properties (breaking length, Young’s modulus, tear index, burst index and elongation) and are displayed in Figure 2.

The breaking length of the industrial paper in the machine direction has a value of 5656 m. Considering the anisotropy/isotropy ratio of 1.65, the breaking length of sheets made in isotropic former should be 3428 m. The Ref. sample shows a breaking length of 1895 m, so a significant decrease in mechanical properties due to the hornification process was observed. In general, it was observed that the application of the nanofibers obtained in this work as a reinforcing agent in recycled linerboard considerably improved the mechanical properties of the sheets obtained. Among the samples obtained, only the incorporation of HR-TCNF and VS-TCNF managed to counteract the loss of mechanical properties suffered, giving rise to 3429 and 3923 m, respectively. So, in this sense, recycled linerboard fibers reinforced with HR-TCNF and VS-TCNF would make it possible to obtain recycled linerboard with the characteristics of industrial linerboard. A greater specific surface area of the nanofibers, as in the case of HR-TCNF and VS-TCNF, leads to greater interaction between adjacent fibers resulting in a reinforcing effect. The nanofibers act as promoters of fiber–fiber adhesion, generating an embedded network between the matrix that contributes to increase the load-bearing capacity of the linerboard [1,34]. Similar improvements have been reported in literature when using LCNF from different raw materials such as orange tree pruning or pine sawdust [1,35]. In both cases, the breaking length improvements reached 30–40% compared to Ref. sample. No significant differences were observed in breaking length comparing both HR- and VS-derived nanofibers or the lignin content in the substrates. Comparing the data obtained in the best cases with the mechanical beating data, it was observed that, in both cases, the breaking length value was exceeded. Mechanical beating resulting in a breaking length of 3139 and 3400 m for 2000 and 3000 rev, respectively. As in industrial recycling, sufficient mechanical refinement is able to counteract the loss of mechanical properties of recycled paperboard. The use of HR-TCNF or VS-TCNF resulted in an improvement of 15% over mechanical beating. These data prove the technical feasibility of CNFs obtained from agricultural residues as a technology to be used in linerboard recycling processes, achieving a higher reinforcement effect and even extending the number of recycling cycles of the products due to the non-physical modification of the fibers.

In the case of Young’s modulus, the improvement of the linerboard sheets by adding the nanofibers was evident, following the same trend as that for breaking length. In the case of HR nanofibers, the best results for Young’s modulus were obtained with HR-LCNF and HR-TCNF. In both cases, this improvement was more than 220%. When VS nanofibers were used, the best results for Young’s modulus were achieved when using VS-CNF and VS-TCNF. It seems that, in this case, the pretreatment applied to obtain nanofibers had no influence, but the absence of lignin was decisive in obtaining optimum results. For this raw material, the improvement achieved was also almost 240%. This improvement in mechanical properties is due to the high specific surface area of the fibers introduced into the recycled linerboard, which will also produce a higher swelling capacity. These facts result in a higher number of inter-fiber bonds per unit, increasing the relative bond area. Mechanical beating resulted in 1.15 and 1.43 GPa for 2000 and 3000 rev, respectively. The use of HR and VS nanofibers obtained by any pretreatment and regardless of lignin content, outperformed the mechanical beating values at 2000 rev. The beating at 3000 rev was only surpassed in the case of the best reinforcements, as explained above. These data again corroborate the suitability of agricultural residue-derived nanofibers for use in this type of process as a substitute for mechanical beating. Several authors have demonstrated the suitability of various agricultural residues such as bamboo or oil palm tree for the preparation of nanofibers and their subsequent incorporation into composite materials by improving Young’s moduli [36,37].

Tear index showed no major improvements with respect to recycled linerboard or the ones with the fibers subjected to mechanical beating, except for TCNF. As occurred for the breaking length, HR-TCNF and VS-TCNF improved the tear index by 27% over the mechanical beating. Regarding the burst index, only in the case of VS-TCNF were the values of mechanical beating slightly exceeded. This index describes the intensity of the material when resisting a disc-shaped diaphragm that attempts to burst it. Properties such as fiber length, fiber quality, internal sizing and surface treatment also influence the burst index of the final material [38]. The data obtained here are similar to those presented for linerboard incorporating 3% wt. nanofibers from never-dried Northern bleached softwood kraft pulp [11]. Finally, significant improvements were achieved in elongation in the case of chemically pretreated nanofibers obtained from both raw materials. The high specific surface area of nanofibers obtained by chemical pretreatment, as in the case of HR-TCNF and VS-TCNF, underlies this overall improvement in mechanical performance, making them competent with industrially applied mechanical beating.

Another parameter to be considered when applying reinforcement treatments in the recycling processes of paper production is the drainability of the slurries. This parameter is indicative of the capacity of a given pulp to retain water and, therefore, its value affects the production rate of a given paper or linerboard machine. Figure 3 shows the drainability of the slurries obtained in this study through the Schopper–Riegler index. The Schopper–Riegler index (°SR) is the freeness test performed to measure the drainability of pulps. When different pulps are obtained by pulping and refining at similar conditions, the drainability will depend mostly on the raw material and the percentage of fines present in the pulp. In general, the increase in pulp refining results in a higher number of fines, which translates into an increase in °SR [39]. It was observed that nanofiber reinforcement increased the °SR values negatively affecting the drainability of the slurries. This is mainly due to the high water-holding capacity of the nanofibers (due to their high anionic charge) and their nanometric size which strongly reduces the porosity of the wet web, thus hindering water permeation. Taipale et al. (2010) and Hii et al. (2012) observed that drainage time increased proportionally to CNF does due to blockage of leaf pores during their formation [40,41]. In the present work, it was observed that nanofibers obtained by mechanical pretreatment presented a much lower increase in water-holding capacity than those obtained by chemical pretreatment. This is due to the higher anionic charge and smaller diameter of the nanofibers obtained by TEMPO-mediated oxidation. A significant difference in this property is also observed with respect to the presence of lignin in the nanofibers. A residual lignin content in the fiber increases the hydrophobicity of the fibers, thus decreasing the water-holding capacity and significantly increasing the drainability of the fibers, showing lower °SR values. Although the negative effect of the addition of the nanofibers on drainage is proven, drainage can be maintained or even improved when an adequate retention system (RS) is used. The most used RSs are those that use the combination of polymers and microparticles to control chemical flocculation and optimize retention, drainage and formation. Several authors have observed that the use of polyelectrolytes such as polyacrylamide (PAM), or cationic polyacrylamide (CPAM), when using cellulose nanofibers as a reinforcing agent, improves load retention and mechanical properties without a detrimental effect on drainage [8,42]. It is also possible to use dual RSs composed of CPAM and bentonite to improve the drainage of suspensions without affecting the improvement in mechanical properties [43].

The results obtained show the suitability of unexplored agricultural residues such as horticultural residues and vine shoots to produce nanofibers with suitable characteristics for their application as a reinforcing agent in linerboard recycling process. This production of nanofibers can be carried out by mechanical or chemical pretreatment, depending on the costs to be assumed. Moreover, there will be no need for the removal of the lignin present, thus reducing process costs due to the elimination of bleaching steps and reducing the environmental impact. As discussed in previous works, the high mechanical strength obtained in this work allows the adoption of several market strategies. The first is to reduce the basis weight or increase the mineral filler content, thus reducing the consumption of fibrous raw materials and, therefore, the costs of the process. The next would be the introduction of these materials in market niches with higher technical requirements [44]. Prior to all this, the application of nanofibers as a reinforcing agent in the linerboard recycling process must be analyzed from a technical and economic point of view. The economic study of different pretreatments and treatments has been a topic of special interest in the production of nanofibers. The cost of the different treatments is mainly due to the energy consumption during mechanical beating on the one hand, and to the consumption of reagents and catalyst in the case of TEMPO-mediated oxidation. In a previous work, we have reported that there are considerable differences between the production costs of both pretreatments [1]. This is due to the high price of the TEMPO catalyst, which raises the production cost of nanofibers by more than 80 fold. Therefore, the use of chemically pretreated nanofibers for this application will be an alternative to evaluate when the recovery and reuse of the catalyst is possible. The use of mechanically pretreated nanofibers can be a good alternative, but more studies on the incorporation of larger quantities of these nanofibers in the linerboard are needed to reach the values offered by TEMPO nanofibers. In addition, the use of other treatments such as twin-screw extruders to replace HPH can lead to a great reduction in the energy consumption of the nanofibrillation treatment, making this technology more competitive [45].

## 4. Conclusions

Tomato, eggplant and bell pepper horticultural residues and vine shoots have been used to produce cellulose nanofibers by different pretreatments and from cellulose pulps with different compositions. The following conclusions are drawn from this work:
Both HR and VS resulted in nanofibers with good properties for application as reinforcement in linerboard sheets.The most remarkable nanofibers were HR-TCNF and VS-TCNF, which presented a cationic demand of 1043.54 and 1227.91 µeq·g/g, a carboxyl content of 148.12 and 168.93 µeq·g/g, and a specific surface area of 436 and 516 m^2^/g, respectively, in addition to nanofibrillated fractions of more than 60%.The use of the nanofibers obtained as reinforcement in the linerboard resulted in an improvement of the mechanical properties of the recycled linerboard. The most significant increase was obtained when using HR-TCNF and VS-TCNF, which offset the loss of mechanical properties suffered during recycling. However, these samples were the ones that greatly reduced the drainability of the suspensions.It was concluded that the use of cellulose nanofibers as a reinforcing agent outperforms the mechanical reinforcement produced by conventional mechanical refining used in the industry and extends the number of recycling cycles of the products due to the non-physical modification of the fibers.

## Figures and Tables

**Figure 1 nanomaterials-12-04447-f001:**
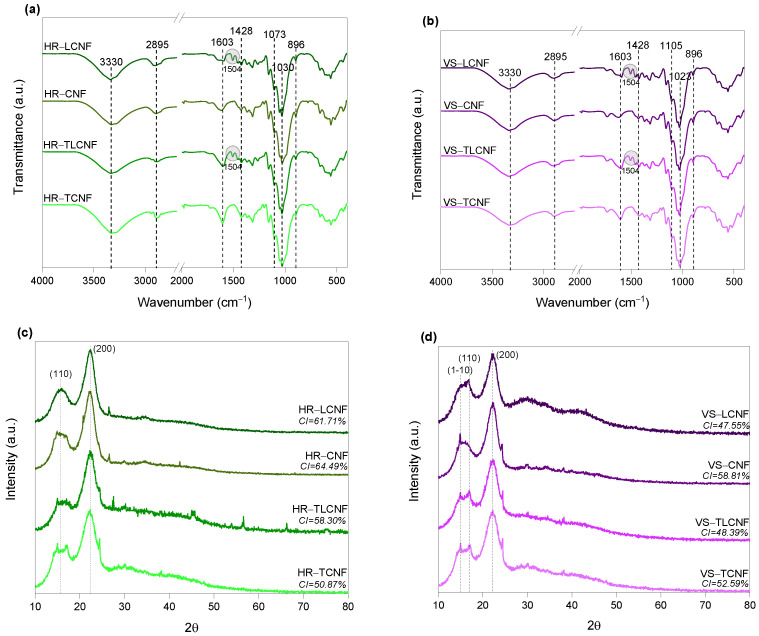
FTIR spectra of the nanofibers obtained from (**a**) horticultural residues and (**b**) vine shoots; X-ray diffraction patterns of the nanofibers obtained from (**c**) horticultural residues and (**d**) vine shoots.

**Figure 2 nanomaterials-12-04447-f002:**
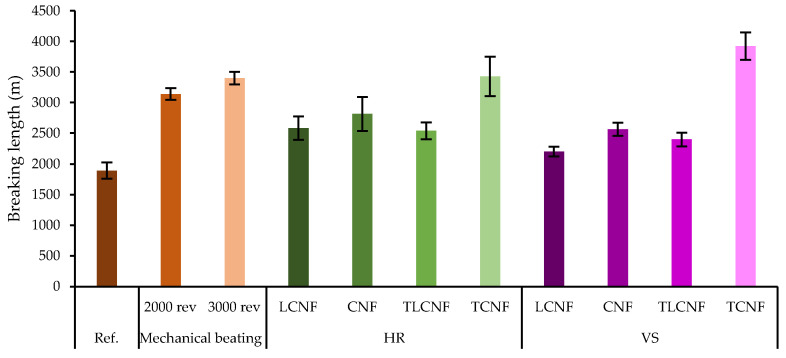
Mechanical properties of recycled linerboard with different nanofibers (3% on fiber dry weight) and treatments.

**Figure 3 nanomaterials-12-04447-f003:**
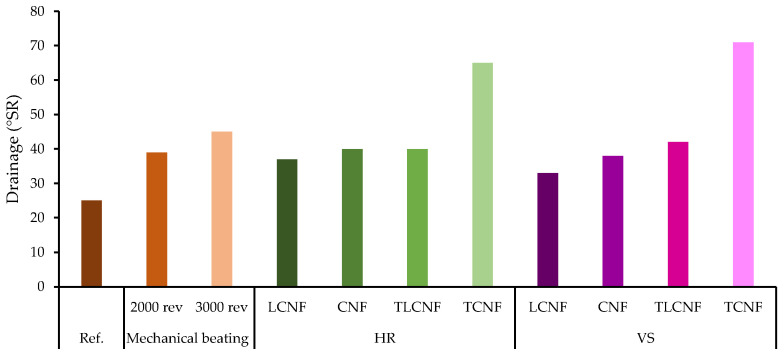
Drainage capacity evolution.

**Table 1 nanomaterials-12-04447-t001:** Nomenclature for the nanofibers obtained in this work.

Sample Name	Origin (Harvesting Residue)	Cellulose Pulp	Pretreatment (Prior to Nanofiber Obtention)
HR-LCNF	Horticultural residues	Unbleached	Mechanical
HR-CNF		Bleached	
VS-LCNF	Vine shoots	Unbleached	
VS-CNF		Bleached	
HR-TLCNF	Horticultural residues	Unbleached	Chemical
HR-TCNF		Bleached	
VS-TLCNF	Vine shoots	Unbleached	
VS-TCNF		Bleached	

**Table 2 nanomaterials-12-04447-t002:** Chemical composition of the agricultural residues and their cellulose pulps, determined according to TAPPI standards [17].

	Extractives (%)	Ashes (%)	Lignin (%)	Hemicelluloses (%)	α-Cellulose (%)
HR ^1^	17.12 ± 1.55	14.86 ± 1.35	15.03 ± 1.36	20.89 ± 1.90	32.10 ± 2.91
UBP-HR ^2^	11.02 ± 0.86	4.68 ± 0.02	16.00 ± 0.63	18.17 ± 1.21	50.45 ± 2.81
BP-HR ^3^	9.11 ± 0.47	3.20 ± 0.08	7.31 ± 0.51	16.23 ± 1.18	64.42 ± 7.55
VS ^4^	16.84 ± 2.38	1.96 ± 0.76	19.98 ± 1.89	22.06 ± 1.48	39.17 ± 0.35
UBP-VS ^5^	16.97 ± 2.00	3.49 ± 0.17	14.20 1.09	13.54 ± 0.97	51.89 ± 1.04
BP-VS ^6^	8.96 ± 0.41	3.27 ± 0.35	6.42 ± 2.37	12.43 ± 0.44	67.17 ± 5.91

^1^ Horticultural residues and its ^2^ unbleached and ^3^ bleached pulp; ^4^ vine shoots and its ^5^ unbleached and ^6^ bleached pulp.

**Table 3 nanomaterials-12-04447-t003:** Characterization of (ligno)cellulose nanofibers.

	Nanofibrillated Fraction (%)	Optical Transmittance (%)	Cationic Demand (µeq·g/g)	Carboxyl Content (µeq·g/g)	Specific Surface Area (m^2^/g)	Length (nm)	Diameter (nm)
HR-LCNF	23.63 ± 4.14	23.9	301.28 ± 28.39	35.69 ± 6.53	129	1536	19
HR-CNF	38.22 ± 6.46	35.5	688.24 ± 18.90	77.28 ± 17.11	298	4960	8
HR-TLCNF	49.52 ± 2.95	44.3	621.55 ± 9.45	83.83 ± 4.28	262	813	9
HR-TCNF	63.44 ± 4.52	74.7	1043.54 ± 18.2	148.12 ± 5.26	436	614	6
VS-LCNF	18.42 ± 2.73	20.1	398.67 ± 18.79	31.49 ± 8.40	179	1418	14
VS-CNF	33.35 ± 3.50	35.7	407.44 ± 9.46	88.94 ± 11.25	155	6936	16
VS-TLCNF	45.05 ± 2.88	35.4	744.60 ± 9.49	80.85 ± 15.89	323	1095	8
VS-TCNF	60.42 ± 5.56	62.2	1227.91 ± 18.8	168.93 ± 10.25	516	755	5

## Data Availability

The data presented in this study are available on request from the corresponding author.

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
