# Peer review of "Nanocellulose from Spanish Harvesting Residues to Improve the Sustainability and Functionality of Linerboard Recycling Processes"

_nanomaterials, 2022, doi:10.3390/nano12244447_

Round 1

Reviewer 1 Report

In this study, the authors investigated the viability of two different agricultural residues to produce LCNF and CNF, and evaluated different pretreatment methods to produce the nanofibers with good properties. It is suggested to be published after making the following modifications.

1. Please check the spelling format of line no.188 to keep the names uniform.

2. The length and diameter of cellulose nanofibers are listed in Table 2, and it is suggested to put the surface morphology photos of cellulose nanofibers.

3. Is there a recovery process for the wastewater after reaction? Does it reduce air and water pollution compared to the production of new paper or cardboard?

Author Response

In this study, the authors investigated the viability of two different agricultural residues to produce LCNF and CNF and evaluated different pretreatment methods to produce the nanofibers with good properties. It is suggested to be published after making the following modifications.

  1. Please check the spelling format of line no.188 to keep the names uniform.

According to the reviewer’s comment, the spelling format of this line has been corrected.

  1. The length and diameter of cellulose nanofibers are listed in Table 2, and it is suggested to put the surface morphology photos of cellulose nanofibers.

As indicated by the reviewer the length and diameter of cellulose nanofibers was defined in this article. Morphology of the nanofibers can be determined by SEM (or other techniques for direct observation). However, these techniques have certain disadvantages when used. This fact entails special difficulty due mainly to the aggregation of nanofibers during the drying of the sample. Several recent articles have focused on this topic, optimizing the processes of characterization by microscopy so that these methods are effective and reproducible:

  • https://doi.org/10.1007/s10570-020-03138-1
  • https://doi.org/10.1007/s10570-020-03116-7

However, due to the limitations exposed above, the calculation of a mean diameter value representative of the whole sample by SEM observation is not appropriate. Therefore, the authors maintain in the Table the value obtained by the theoretical model described in the article “Considering the stoichiometric relationship between hydroxyl and carboxyl groups with poly-DACMAC, the specific surface area (σ) and diameter (d) of the nanofibers were estimated, following previously published methodologies [3], according to equations (1) and (2), respectively.“ which effectiveness has been demonstrated in several manuscripts e.g. (M. Delgado-Aguilar, I. González, Q. Tarrés, M. Àngels Pèlach, M. Alcalá, The key role of lignin in the production of low-cost lignocelllosic nanofibres for papermaking applications, Ind Crop Prods. 86 (2016) 295-300 ; E. Espinosa, Q. Tarrés, M. Delgado-Aguilar, I. González, P. Mutjé, A. Rodríguez, Suitability of wheat straw semichemical pulp for the fabrication of lignocellulosic nanofibers and their application to papermaking slurries, Cellulose. 23 (2016) 837-852) with deviation of ±0.8 nm with respect to the diameter studied by microscopy. The length has also been determined by theorical approach as described in the manuscript: “The degree of polymerization (DP) was determined from the intrinsic viscosity measurements, according to UNE 57-039-92 [20]. The DP data was used to calculate the length of the fibers according to the equation (3), proposed by Shinoda et al. [21].” Due to the low representativeness of the microscope observations, and the lack of time in the present review, the authors consider that it is not vital to take these photographs, since they do not improve the results obtained.

  1. Is there a recovery process for the wastewater after reaction? Does it reduce air and water pollution compared to the production of new paper or cardboard?

After the soda pulping process to which the raw materials are subjected to obtain the cellulose pulp, the “wastewater” is recovered as black liquor, where the lignin is dissolved. This black liquor is valorized through other processes, including the precipitation and purification of the lignin for its application in the preparation of new functional materials. In this way, it is intended to carry out an integral utilization of the harvesting residues, valorizing all the lignocellulosic fractions present.

Regarding the reduction in air and water pollution, Miao et al., (2018)1 have reported that the production of each ton of recycled paper using secondary fibers saves 1.2 tons of standard coal, 600 kWh of electric power, 100 m3 of water, and 300 kg of raw chemical materials, reduces the deforestation of 1.7 x 103 m3 of forest, landfill usage by three cubic meters, air pollution by 74%, and water pollution by 35% compared to the production of new virgin paper. It is these data that are referred to in the Introduction section of the manuscript (lines 46-49) when it is stated that the production of cardboard with secondary fibers reduces pollution.

1 https://doi.org/10.1515/npprj-2018-0014

Reviewer 2 Report

This work presents an experimental study of the influence of several nanofibers as a reinforcing agent in the recycled cardboard.

The nanofibers are added to improve the mechanical properties. The effect of the nanofibers on mechanical behavior is evaluated with Breaking length, Young’s modulus, Tear Index, Burst Index and Elongation tests.

The methodology of the study is clear, the paper is well written and well organized and the original contribution is well explained. Only minor revision is necessary before publication.

1)      out of 44 citations reported in the bubliography there are up to 13 self-citations by Espinoza. etc.

The number of self-citations is too high. it is suggested to limit their use only to those necessary.

Author Response

This work presents an experimental study of the influence of several nanofibers as a reinforcing agent in the recycled cardboard.

The nanofibers are added to improve the mechanical properties. The effect of the nanofibers on mechanical behavior is evaluated with Breaking length, Young’s modulus, Tear Index, Burst Index and Elongation tests.

The methodology of the study is clear, the paper is well written and well organized and the original contribution is well explained. Only minor revision is necessary before publication.

 1)      out of 44 citations reported in the bubliography there are up to 13 self-citations by Espinoza. etc.P

The number of self-citations is too high. it is suggested to limit their use only to those necessary.

Authors thanks the reviewer for his careful revision of the manuscript. The number of self-citation has been considerable decreased.

Reviewer 3 Report

This seems to be a well-done study with some interesting results.  A few concerns and suggestions are noted below.   While the TCNF materials gave the best reinforcement, they likely will be quite a bit more expensive.  The authors only report the results at 3% addition of the CNF to the pulp fibers.  Maybe the mechanical only produced materials could give similar properties if added at a bit higher amount.   The authors do not say why they decided on 3% addition.  This can be a good publication after modifications. 

1.        In abstract define abbreviations or do not use them.  HR-TCNF and VS-TCNF

2.      I do not think this statement is true  “one ton of standard coal,”.   Using recycled fibers for sure saves trees and chemicals, but because the drying is from non-biomass source, it may use more energy.  Anyway, this statement is not needed in the paper and should be removed or a reference provided.

3.      Table 2.  Instead of foot notes defining the various columns, why not just label them?   Significant figures do not seem correct here.

4.      I do not understand the word “nanofibrillation”.   A small amount of fibrillation?

5.      Maybe I am missing this in the text, but the method to characterize the fiber length and diameter is not described.   

6.      Page 6.  I am not sure the word “yield” is good when talking about fibrillation.  It implies that you are losing material when you are producing the nano-scale fibers.  Maybe the authors could say something like the quantity of material that becomes fine enough to not scatter light.

7.      The couple of sentences:  Regarding the specific surface area, all the samples obtained in this work reach interesting values for their application as cardboard reinforcement. A higher specific surface allows a greater capacity of interaction with the surrounding fibers, thus increasing the fiber-fiber adhesion and favoring the reinforcing effect of the final product. Are not obvious.  What are “interesting” values?  It is not clear that finer materials will do better that microns sized material. 

8.      Figure 2.  What content of the CNF or similar is used here?  Please report in the figure caption. 

9.      The word “Carboard” implies the fluted structure of two liners and a fluted center.  They are testing linerboard. 

10.   The TCNF material should be made clear that this could be expensive in some manner.  Yes, it is the most “remarkable” material but to use this to reinforce paperboard likely is not going to make economic sense. 

11.   The discussion around drainage maybe can be improved.   Likely the poor drainage is more a result of the fibers blocking pathways of water removal over what is discussed.   The solution to poor drainage is likely not as easy to find as the authors indicate. 

12.   In the conclusions, again I would modify the word “yield”. 

13.   In the conclusions, yes the TCNF gave the best reinforcement but also the poor drainage.  They should note this result in the conclusions as well.   

Author Response

This seems to be a well-done study with some interesting results.  A few concerns and suggestions are noted below.   While the TCNF materials gave the best reinforcement, they likely will be quite a bit more expensive.  The authors only report the results at 3% addition of the CNF to the pulp fibers.  Maybe the mechanical only produced materials could give similar properties if added at a bit higher amount.   The authors do not say why they decided on 3% addition.  This can be a good publication after modifications. 

  1. In abstract define abbreviations or do not use them.  HR-TCNF and VS-TCNF.

The abbreviations in the abstract have been avoided.

  1. I do not think this statement is true  “one ton of standard coal,”.   Using recycled fibers for sure saves trees and chemicals, but because the drying is from non-biomass source, it may use more energy.  Anyway, this statement is not needed in the paper and should be removed or a reference provided.

According to the reviewer’s comment, this statement has been rewritten keeping the thread of this part of the Introduction section.  

  1. Table 2.  Instead of foot notes defining the various columns, why not just label them?   Significant figures do not seem correct here.

The foot notes of the table have been deleted and the name of the columns have been defined.

  1. I do not understand the word “nanofibrillation”.   A small amount of fibrillation?

Over the past twenty years, the term “nano” has become widely used. This term refers to structures having at least one dimension between approximately 1.0 and 100 nm. Regarding the name given to the material composed of cellulose fibrils, evidence provided in the literature based, for example, on microscopy and image analysis suggest that the products derived from fiber disintegration are somewhat inhomogeneous in size, but most of the material appears to be composed of fibrils with diameters less than 100 nm (see, for example, https://doi.org/10.1007/s10570-009-9290-4; https://doi.org/10.1016/j.carbpol.2020.116186; https://doi.org/10.1016/j.indcrop.2019.01.046). Thus, nanoscale-related terms are justified. Therefore, to be consistent with the modern definitions of various scales (e.g., nanometer, submicrometer, and micrometer), in this manuscript the term “nanofibrillation” is applied to refer to the process by which pulp cellulose fibers defibrillate to smaller components, reaching the nanoscale size.

  1. Maybe I am missing this in the text, but the method to characterize the fiber length and diameter is not described.   

In the Materials and Methods section, the published works where the equations used for the estimation of the length and diameter of the nanofibers are found are referenced. For clarity in this regard, the equations have also been included in this manuscript.

  1. Page 6.  I am not sure the word “yield” is good when talking about fibrillation.  It implies that you are losing material when you are producing the nano-scale fibers.  Maybe the authors could say something like the quantity of material that becomes fine enough to not scatter light.

According to the reviewer’s suggestions, in order not to be misleading, the term “yield” has been replaced by “fraction” referring to the amount of material that has reached the nanometer size and does not scatter light.

  1. The couple of sentences:  Regarding the specific surface area, all the samples obtained in this work reach interesting values for their application as cardboard reinforcement. A higher specific surface allows a greater capacity of interaction with the surrounding fibers, thus increasing the fiber-fiber adhesion and favoring the reinforcing effect of the final product. Are not obvious.  What are “interesting” values?  It is not clear that finer materials will do better that microns sized material. 

The authors thank the reviewer for his annotation. The sentence has been changed to improve its understanding. The higher effect of nanofibers versus microfibrillated materials has been reported several times in literature. Delgado-Aguilar, Gonzalez et al. (2015) prepared CNFs with a high-pressure homogenizer using different pretreatment methods (chemical, mechanical and enzymatic). The resulting CNFs presented different fibrillation yields and specific surface area depending on the type of pretreatment. These CNFs were added to bleached hardwood kraft pulp and tensile testing of the resulting paper revealed that the best improvements in strength were observed in CNFs with high fibrillation yields (>90%). However, the authors also demonstrated that CNFs with lower fibrillation yield and, consequently, lower energy cost, are also capable of producing a significant increase in tensile strength, although they did not reach such high values. In a similar experiment, Su, Mosse, Sharman, Batchelor, and Garnier (2013) also compared the effect of cellulose microfibrils (CMPs) prepared by cryogenic milling and commercial nanofibers. Nanoscale CNFs produced better improvement in tensile strength and density compared to their micrometer counterpart.

  • Su, W.K.J. Mosse, S. Sharman, W.J. Batchelor, G. Garnier. Effect of tethered and free microfibrillated cellulose (MFC) on the properties of paper composites Cellulose, 20 (2013), pp. 1925-1935
  • Delgado-Aguilar, I. González, Q. Tarrés, M. Alcalá, M.A. Pèlach, P. Mutjé. Approaching a low-cost production of cellulose nanofibres for papermaking applications. BioResources, 10 (3) (2015), pp. 5345-5355

  1. Figure 2.  What content of the CNF or similar is used here?  Please report in the figure caption. 

The concentration of the nanofibers used for the preparation of the cardboard sheets was 3% on fiber dry weight. This information has been included in the caption of Figure 2.

  1. The word “Carboard” implies the fluted structure of two liners and a fluted center.  They are testing linerboard. 

The authors agree with the reviewer. The word cardboard was originally used to make the material used more generic and well-known. The term 'cardboard' has been replaced by 'linerboard' for greater specificity of the material used.

  1. The TCNF material should be made clear that this could be expensive in some manner.  Yes, it is the most “remarkable” material but to use this to reinforce paperboard likely is not going to make economic sense. 

Authors agree with the reviewer in this regard. A brief discussion about the economic viability of the work has been added at the end of the Results and discussion section (lines 437-454): Prior to all this, the application of nanofibers as a reinforcing agent in the cardboard recycling process must be analyzed from a technical and economic point of view. The economic study of different pretreatments and treatments has been a topic of special interest in the production of nanofibers. The cost of the different treatments is mainly due to the energy consumption during mechanical beating on the one hand, and to the consumption of reagents and catalyst in the case of TEMPO-mediated oxidation. In a previous work, we have reported that there are considerable differences between the production costs of both pretreatments [1]. This is due to the high price of the TEMPO catalyst, which raises the production cost of nanofibers by more than 80 times. Therefore, the use of chemically pretreated nanofibers for this application will be an alternative to evaluate when the recovery and reuse of the catalyst is possible. The use of mechanically pretreated nanofibers can be a good alternative, but more studies on the incorporation of larger quantities of these nanofibers in the cardboard are needed to reach the values offered by TEMPO-nanofibers. In addition, the use of other treatments such as twin-screw extruders to replace HPH can lead to a great reduction in the energy consumption of the nanofibrillation treatment, making this technology more competitive [47].”

  1. The discussion around drainage maybe can be improved.   Likely the poor drainage is more a result of the fibers blocking pathways of water removal over what is discussed.   The solution to poor drainage is likely not as easy to find as the authors indicate. 

The authors appreciate the comment and have modified and improved the discussion of suspension drainage. “This is mainly due to the high-water holding capacity of nanofibers (due to their high an-ionic charge) and their nanometric size which strongly reduces the porosity of the wet web, thus hindering water permeation. Taipale et al. and Hii et al. ob-served that drainage time increased proportionally to CNF doses due to blockage of leaf pores during their formation. In the present work, it is observed that nanofibers obtained by mechanical pretreatment presented a much lower increase in water holding capacity than those obtained by chemical pretreatment. This is due to the higher anionic charge and smaller diameter of the nanofibers obtained by TEMPO-mediated oxidation. A signif-icant difference in this property is also observed with respect to the presence of lignin in the nanofibers. A residual lignin content in the fiber increases the hydrophobicity of the fibers, thus decreasing the water holding capacity and significantly increasing the drain-ability of the fibers, showing lower °SR values. Although the negative effect of the addition of nanofibers on drainage is proven, drainage can be maintained or even improved when an adequate retention system (RS) is used. The most used retention systems are those that use a combination of polymers and microparticles to control chemical flocculation and optimize retention, drainage and formation. Several authors have observed that the use of polyelectrolytes such as polyacrylamide (PAM), or cationic polyacrylamide (CPAM), when using cellulose nanofibers as a reinforcing agent, improves load retention and me-chanical properties without a detrimental effect on drainage.  It is also possible to use dual retention systems composed of CPAM and bentonite to improve the drainage of suspensions without affecting the improvement in mechanical properties.”

  1. In the conclusions, again I would modify the word “yield”.

As stated in point 6, the term “yield” has been replaced by “fraction”.  

  1. In the conclusions, yes the TCNF gave the best reinforcement but also the poor drainage.  They should note this result in the conclusions as well.  

This fact have been added to conclusions. “3.    The use of the nanofibers obtained as reinforcement in the linerboard resulted in an improvement of the mechanical properties of the recycled linerboard. The most significant increase was obtained when using HR-TCNF and VS-TCNF, which offset the loss of mechanical properties suffered during recycling. However, these samples were the ones that greatly reduced the drainability of the suspensions.”

Reviewer 4 Report

The research of De-Haro Niza et al is focused on paper recycling. In particular, given the critical issues deriving from typical hornification processes in terms of reduction of the mechanical properties of the products, the authors demonstrate how the exploitation of hitherto unexplored agricultural residues can allow the production of nanofibers applicable as reinforcing agents in the recycling process of cardboard.

Very interesting work from a scientific point of view and in line with the themes of circular economy and sustainability but often expressed in a hermetic language that can only be understood by insiders.

In other words, while appreciating the appropriateness of the procedures adopted and the discussion of the results obtained, MINOR REVISIONS are required to clarify the meaning of some abbreviations, at least the first time they are inserted in the text. Specifically, please specify the codes: TCNF, TLCNF, TEMPO.

Greater clarity is also recommended for the so-called "burst index" and “Schopper-Riegler degree” terms: parameters probably not known to all readers who generally deal with materials engineering.

Author Response

The research of De-Haro Niza et al is focused on paper recycling. In particular, given the critical issues deriving from typical hornification processes in terms of reduction of the mechanical properties of the products, the authors demonstrate how the exploitation of hitherto unexplored agricultural residues can allow the production of nanofibers applicable as reinforcing agents in the recycling process of cardboard.

Very interesting work from a scientific point of view and in line with the themes of circular economy and sustainability but often expressed in a hermetic language that can only be understood by insiders.

In other words, while appreciating the appropriateness of the procedures adopted and the discussion of the results obtained, MINOR REVISIONS are required to clarify the meaning of some abbreviations, at least the first time they are inserted in the text. Specifically, please specify the codes: TCNF, TLCNF, TEMPO.

According to the reviewer’s comment, a new table (Table 1) has been inserted in the manuscript displaying the meaning of the sample names and abbreviations used throughout the manuscript. The meaning of “TEMPO” is explained in the Materials section.

Greater clarity is also recommended for the so-called "burst index" and “Schopper-Riegler degree” terms: parameters probably not known to all readers who generally deal with materials engineering.

The description of the Burst index parameter has been added in lines 391-394: Regarding the Burst index, just in the case of VS-TCNF the values of mechanical beating were slightly exceeded. This index describes the intensity of the material when resisting a disc-shaped diaphragm that attempts to burst it.  Properties such as fiber length, fiber quality, internal sizing and surface treatment also influence the burst index of the final material”.  

Similarly, the description of the Schopper-Riegler degree has been included in lines 404-410: Figure 3 shows the drainability of the slurries obtained in this study through the Schopper-Riegler index. The Schopper-Riegler index (°SR) is the freeness test performed to measure the drainability of pulps. When different pulps are obtained by pulping and refining at similar conditions, the drainability will depend mostly on the raw material and the percentage of fines present in the pulp. In general, the increase in pulp refining results in a higher number of fines, which translates into an increase in °SR”.

Reviewer 5 Report

The article :Nanocellulose from Spanish harvesting residues to improve the sustainability and functionality of cardboard recycling processes" is interesting. However, it requires correction before being accepted for publication.

- Please add the numerical values of the improvement of the obtained properties in the abstract and please explain the abbreviations used: HR-TCNF and VS-CNF

-Table 1. - How was the chemical composition determined

-It is worth adding tables with the name of specific samples and their origin, which would be easier for the analysis.(horticultural residues (HR) and vine shoots (VS)

- Please explain the abbreviations UBP-HRBP-HR used, UBP-VS and BP-VS

- Please explain why measuring the drainability of the slurries is so important

- The analysis of mechanical properties needs to be supplemented with a reference to literature data. Because in its present form, we can only see an increase or decrease. no scientific discussion.

Author Response

The article :Nanocellulose from Spanish harvesting residues to improve the sustainability and functionality of cardboard recycling processes" is interesting. However, it requires correction before being accepted for publication.

- Please add the numerical values of the improvement of the obtained properties in the abstract and please explain the abbreviations used: HR-TCNF and VS-CNF

The numerical data of the improvements achieved in the mechanical properties on the cardboard sheets has been included in the abstract. The abbreviations of the sample names have been avoided for better understanding.

-Table 1. - How was the chemical composition determined

The chemical composition displayed in the Table (now named as Table 2) was determined according to TAPPI standards (explained in Materials section). This information has also been included in Table 2 caption.

-It is worth adding tables with the name of specific samples and their origin, which would be easier for the analysis.(horticultural residues (HR) and vine shoots (VS)

According to the reviewer’s comment, a new table (Table 1) has been inserted in the manuscript displaying the meaning of the sample names and abbreviations used throughout the manuscript.

- Please explain the abbreviations UBP-HRBP-HR used, UBP-VS and BP-VS

The explanation of these abbreviations has been included in the new Table 1.

- Please explain why measuring the drainability of the slurries is so important

The importance of carrying out this measurement lies in its industrial application, since a pulp that retains a large amount of water will negatively affect the production rate of a paper or cardboard machine. In this sense, a brief explanation of drainability and Schopper-Riegler degree has been included in the lines 401-410: “Another parameter to be considered when applying reinforcement treatments in the recycling processes of paper production is the drainability of the slurries. This parameter is indicative of the capacity of a given pulp to retain water and, therefore, its value affects the production rate of a given paper or cardboard machine. Figure 3 shows the drainability of the slurries obtained in this study through the Schopper-Riegler index. The Schopper-Riegler index (°SR) is the freeness test performed to measure the drainability of pulps. When different pulps are obtained by pulping and refining at similar conditions, the drainability will depend mostly on the raw material and the percentage of fines present in the pulp. In general, the increase in pulp refining results in a higher number of fines, which translates into an increase in °SR [41].”

- The analysis of mechanical properties needs to be supplemented with a reference to literature data. Because in its present form, we can only see an increase or decrease. no scientific discussion

According to the reviewer’s suggestion, several literature data have been included in the Results and discussion section to compare the data obtained in this work with those reported in literature.

Round 2

Reviewer 5 Report

The authors have introduced the recommended corrections. The manuscript is ready for publication.